# Towards standardization of measuring anxiety and depression: Differential item functioning for language and Dutch reference values of PROMIS item banks

**Ellen B. M. Elsman**[1], **Gerard Flens**[2], **Edwin de Beurs**[3,4], **Leo D. Roorda**[5], **Caroline B. Terwee**[1] *

1 Department of Epidemiology and Data Science, Amsterdam UMC, Amsterdam Public Health Research Institute, Vrije Universiteit Amsterdam, Amsterdam, The Netherlands, 2 Alliance for Quality in Mental Health Care, Utrecht, The Netherlands, 3 Arkin GGZ, Amsterdam, The Netherlands, 4 Clinical Psychology, Faculty of Social Sciences, Leiden University, Amsterdam, The Netherlands, 5 Amsterdam Rehabilitation Research Center | Reade, Amsterdam, The Netherlands

* cb.terwee@amsterdamumc.nl

## Abstract

### Introduction

The outcomes anxiety and depression are measured frequently by healthcare providers to assess the impact of a disease, but with numerous instruments. PROMIS item banks provide an opportunity for standardized measurement. Cross-cultural validity of measures and the availability of reference values are prerequisites for standardized measurement.

### Methods

PROMIS Anxiety and Depression item banks were completed by 1002 representative Dutch persons. To evaluate cross-cultural validity, data from US participants in PROMIS wave 1 were used and differential item functioning (DIF) was investigated, using an iterative hybrid of logistic regression and item response theory. McFadden's pseudo $R^2$-change of 2% was the critical threshold. The impact of any DIF on full item banks and short forms was investigated. To obtain Dutch reference values, T-scores for anxiety and depression were calculated for the complete Dutch sample, and age-group and gender subpopulations. Thresholds corresponding to normal limits, mild, moderate and severe symptoms were computed.

### Results

In both item banks, two items had DIF but with minimal impact on population level T-scores for full item banks and short forms. The Dutch general population had a T-score of 49.9 for anxiety and 49.6 for depression, similar to the T-scores of 50.0 of the US general population. T-scores for age-group and gender subpopulations were also similar to T-scores of the US general population. Thresholds for mild, moderate and severe anxiety and depression were set to 55, 60 and 70, identical to US thresholds.

**Data Availability Statement:** All relevant data are within the article and its Supporting Information files.

**Funding:** The PROMIS Health Organization is a non-profit charitable foundation and the Dutch-Flemish PROMIS National Center is a network of local members of the PHO who are developing or applying PROMIS measures in the Netherlands or Belgium. Both organizations did not provide support in the form of salaries for authors CT and LR, and did not have any additional role in the study design, data collection and analysis, decision to publish, or preparation of the manuscript. The specific roles of these authors are articulated in the 'author contributions' section. The author(s) received no specific funding for this work.

**Competing interests:** CB Terwee and LD Roorda are members of the PROMIS Health Organization and the Dutch-Flemish PROMIS National Center, which aim to improve health outcomes by developing, maintaining, improving, and encouraging the application of PROMIS in research and clinical practice. This does not alter our adherence to PLOS ONE policies on sharing data and materials. The other authors have no conflict of interest.

## Conclusions

The limited number of items with DIF and its minimal impact, enables the use of standard (US) item parameters and comparisons of scores between Dutch and US populations. The Dutch reference values provide an important tool for healthcare professionals and researchers to evaluate and interpret symptoms of anxiety and depression, stimulating the uptake of PROMIS measures, and contributing to standardized outcome measurement.

## Introduction

Symptoms of anxiety and depression are prevalent among patients with various conditions, such as diabetes [1], cancer [2], cardiovascular diseases cardiovascular diseases [3], and numerous mental health disorders [4, 5]. These symptoms are commonly measured by healthcare providers to assess the impact of disease and its treatment. The importance of measuring anxiety and depression is reflected in the widespread inclusion of both outcomes in Standard Sets for major medical conditions by the International Consortium for Health Outcome Measurement (ICHOM) [6]. Currently, anxiety and depression are included in 16 out of 28 Standard Sets, thereby being among the most commonly included outcomes [7].

To assess the outcomes anxiety and depression in patients or the general population, researchers and clinicians use patient-reported outcome measures (PROMs) [8–10]. Numerous PROMs assessing anxiety or depression exist, although not all of them meet the standards for reliability, validity and feasibility [11–13]. Uniformity in PROMs to measure anxiety and depression is lacking, which makes it difficult to compare their scores, and hinders benchmarking and quality of care improvements. Moreover, it is labor-intensive and costly to build in different PROMs and their scoring algorithms in electronic health records, it is difficult for healthcare providers to use different PROMs for patients with different conditions and interpret the results correctly, and it is burdensome for patients with multiple conditions to complete different PROMs measuring the same construct, which scores are not shared between healthcare providers [14–16].

To work towards standardized outcome measurement of anxiety and depression and to overcome the above mentioned challenges, the Patient-Reported Outcomes Measurement Information System (PROMIS)Ⓡ initiative might provide opportunities (see [17, 18] for an overview of PROMIS and early aims and findings about the initiative). PROMIS aims to develop and maintain a state-of-the-art assessment system to measure patient-reported health with highly accurate, precise and short measures [17, 18]. The PROMIS initiative has resulted in a wide range of universal applicable (generic) item banks for use across patient populations, targeting various constructs, including item banks for measuring anxiety and depression [19–21]. PROMIS item banks can be used to create fixed questionnaires with a small number of items (also known as short forms), or used as a computerized adaptive test (CAT), which is more dynamic [22, 23]. In a CAT, items are selected from an item bank based on a persons' responses. The administration of items stops when a pre-specified criterion is met. As a result, the administration burden is reduced, with a negligible loss of precision.

The PROMIS v1.0 Anxiety item bank contains 29 items [24], whereas the v1.0 Depression item bank contains 28 items [25]. Both item banks can be applied as short forms or CAT. Fixed length short forms of 4, 6, 7 and 8 items exist for anxiety (i.e. the PROMIS Short Form v1.0 –Anxiety 4a, 6a, 7a and 8a respectively), and of 4, 6 and 8 items for depression (i.e. the PROMIS Short Form v1.0 –Depression 4a, 6a, 8a and 8b respectively) [24, 25]. Short forms

with increasing length result in more precise scores. As such, instruments intended for large scale data collection and comparisons of large groups can be short, whereas instruments intended for obtaining individual scores, diagnosing and comparing small groups should be longer. Moreover, instruments intended to monitor health status over time require more precision and thus need to be longer as well. For all these intended uses, CAT-based assessment is a good option because it combines efficiency and precision [26].

Several studies have compared PROMIS anxiety and depression instruments with legacy measures for anxiety and depression, such as the Patient Health Questionnaire, Beck Depression Inventory, General Anxiety Disorder and Centre for Epidemiological Studies Depression [27–32]. These studies conclude that PROMIS anxiety and depression instruments perform similar to these legacy measures, and can be used to screen and evaluate depression and anxiety in the general population, as well as in patient groups [27–32].

PROMIS instruments have been implemented in various institutions and health disciplines, such as orthopedics [33–35], oncology [36] and diabetes [37]. Major translation efforts have been conducted [28, 38–41], including the translation of 17 adult item banks into Dutch-Flemish [42]. The Dutch-Flemish PROMIS item banks for anxiety and depression have been validated in a representative sample of the Dutch general population as well as a clinical sample with common mental disorders [43–45], and they can be used in clinical practice and research involving the Dutch population.

In order to pursue standardized measurement of anxiety and depression and to provide contextual meaning to scores, cross-cultural validity of measures and the availability of reference values are important prerequisites. Cross-cultural validity, by means of differential item functioning (DIF) for language, has not yet been investigated for the Dutch-Flemish PROMIS item banks anxiety and depression [46]. Items should be free of DIF to ensure that the US scoring algorithm, which is the default scoring algorithm by PROMIS convention, is appropriate to use in other countries and that country-specific scores are not biased, in order to compare scores between countries. PROMIS item banks are scaled in such a way that the US general population has a mean score of 50 with a standard deviation of 10 [17]. However, some studies have shown that reference values of PROMIS scales in other countries deviate from the mean score of 50 that is obtained from the US general population [47, 48]. Therefore, this study aims to investigate DIF for language between the Netherlands and the US for the PROMIS Anxiety and Depression item banks, assess its impact, and subsequently provide reference values for these item banks for the Dutch general population.

## Materials and methods

The Medical Ethical Committee of Amsterdam UMC, location VUmc, the Netherlands, confirmed that the study protocol was exempted from ethical approval according to the Dutch Medical Research in Human Subjects Act (WMO), as no experiments were conducted. The study adhered to the tenets of the Declaration of Helsinki.

### Participants and procedures

Data was collected in 2014 [43, 44]. Participants were recruited from an existing internet panel of the Dutch general population by a data collection company (Desan Research Solutions). Participants needed to be representative for the Dutch general population with respect to age distribution, gender, educational level (low, middle, high), region of residence (north, east, south, west) and ethnicity (native Dutch, first- and second-generation western immigrant, first- and second-generation non-western immigrant). Representativeness of participants was compared to data from Statistics Netherlands in 2013 with maximum allowable deviations of

2.5%. Participants were asked to complete the full Dutch-Flemish PROMIS item banks Anxiety and Depression, through a web-based survey in which skipping of items was not allowed. Additionally, participants completed questions regarding their sociodemographic characteristics.

For evaluating DIF for language, data from US participants was obtained from the Health-Measures Dataverse [49], containing PROMIS wave 1 data of 21,113 participants. The calibration subsample of the anxiety and depression item banks was used [21], containing respectively 14,836 and 14,839 respondents.

To investigate how often the DIF items were included in CATs, CATs from an ongoing study in a clinical population sample of adult patients who started outpatient treatment for common mental disorders [32] were assessed.

## Measures

The PROMIS Item Bank v1.0 –Anxiety consists of 29 items that assess self-reported fear (panic, fearfulness), anxious misery (dread, worry), hyperarousal (nervousness, restlessness, tension) and somatic symptoms related to arousal (dizziness, racing heart) [21, 24]. Example items include 'I felt anxious', 'I felt fearful' and 'I felt worried'. The PROMIS Item Bank v1.0 – Depression consists of 28 items that assess self-reported negative mood (guilt, sadness), views of self (worthlessness, self-criticism), social cognition (interpersonal alienation, loneliness) and decreased positive affect and engagement (loss of purpose, meaning and interest) [21, 25]. Example items include 'I felt depressed', 'I felt sad' and 'I felt lonely'. All items have a 7-day recall period and are scored on a 5-point Likert scale with response options 1 = never, 2 = rarely, 3 = sometimes, 4 = often and 5 = always. Total scores are derived from the original US IRT model (i.e. the Graded Response Model (GRM) [50]) and expressed as T-scores, with a mean of 50 and a standard deviation of 10 for the US general population [17]. Higher scores represent more anxiety/depression. In line with PROMIS convention, T-scores were calculated based on the item parameters from the original US calibration sample with expected a posteriori estimates [19]. T-scores can either be calculated by uploading item scores in the online HealthMeasures Scoring Service program, provided by the US Assessment Center [20], or by using the conversion tables in the PROMIS anxiety/depression scoring manuals to convert raw sum scores into T-scores [24, 25]. Scoring Service is the most accurate scoring method available because it uses IRT-based response pattern scoring and can handle missing data (the conversion table can only be used when all items are completed) and was therefore used for obtaining Dutch reference values in this study.

## Statistical analyses

Descriptive statistics were used to summarize sociodemographic characteristics of participants. DIF analyses were conducted with an iterative hybrid of logistic regression and IRT with the lordif package [51] in R. In the logistic regression framework, three regression models were compared: model 1, in which item responses are predicted by the latent trait; model 2, in which item responses are predicted by the latent trait and group (US or NL) membership; and model 3, in which item responses are predicted by the latent trait, group membership (US or NL) and the interaction between these terms. Uniform and non-uniform DIF were assessed by comparing model 1 with model 2 and model 2 with model 3, respectively. The likelihood-ratio $\chi^2$ test with detection criterion R2 was used to detect DIF. McFadden's pseudo $R^2$ was used as a measure of DIF magnitude, with a 2% change being considered as critical threshold [51, 52]. Monte Carlo simulations implemented in the lordif package (1000 replications) were performed to check for type I error inflation [51].

The impact of DIF on item and total scores was assessed by visual inspection of category response curves (CRCs) and test characteristic curves (TCCs) per group. To assess the impact of DIF on short forms and full item bank T-scores, T-scores were calculated with the original US item parameters with expected a posteriori estimates from the GRM model (obtained from HealthMeasures), which is standard practice for PROMIS measures, as well as with a hybrid set of item parameters, and subsequently compared. The hybrid set of item parameters consisted of the original (US) item parameters for the non-DIF items and rescaled Dutch item parameters for the DIF items. Dutch item parameters were obtained by fitting a GRM to the Dutch general population sample, using the mirt package [53] in R. To obtain a hybrid set of item parameters the Stocking-Lord method was used to rescale the Dutch item parameters for DIF items to the US metric [54, 55]. The equate function in the lordif package computes linear transformation constants (with DIF free items as anchor) that can be used to equate the Dutch item parameters to the scale of the US item parameters [51], while minimizing the squared difference between the test characteristic curve. These constants were then used to transform the Dutch discrimination ($\alpha$) and location ($\beta$) parameters of the DIF items into new item parameters ($\alpha_{new}$ and $\beta_{new}$) on the US metric.

The mean T-score of respondents was calculated for the original and hybrid approach, to investigate the impact on T-scores on a group level. Furthermore, for each respondent the absolute difference between the original and hybrid approach was calculated, to investigate the impact on T-scores of individuals. To investigate the impact of DIF on CATs, it was assessed how often the DIF items were included in CATs, based on 4047 CATs for anxiety and 4293 CATs for depression from an ongoing study [32].

To provide reference values for the Dutch general population, T-scores on the complete item banks were calculated with the original US item parameters for the entire group of participants, as well as for age-range (18–34 years, 35–44 years, 45–54 years, 55–64 years, 65–74 years and ≥75 years) and gender subpopulations, in accordance with available subpopulation reference scores of the US population [56]. T-scores of the Dutch general population were compared to the US general population and age-range and gender subpopulation reference scores. T-score ranges that correspond to within normal limits, and to mild, moderate and severe symptoms [57] were computed using thresholds based on mean plus 0.5, 1 and 2 standard deviations. Subsequently, the percentage of participants that would fall within each category was calculated.

## Results

A total of 1486 participants were invited, of which 1055 completed the PROMIS Anxiety and Depression item banks (response rate 71%). Because of suspicious response patterns (e.g. all responses in one category combined with short response times), 53 participants were excluded from the analysis. Sociodemographic characteristics of the remaining 1002 participants are presented in Table 1. Differences in sociodemographic characteristics between the study participants and the Dutch general population in 2013 were all less than 2.5%, except for ethnicity.

Monte Carlo simulations indicated that the type I error rate for DIF detection was well controlled, as the empirical thresholds for probability associated with the $\chi^2$ statistic were all close to the nominal $\alpha$ (= 0.01) level, ranging from 0.009–0.01 for both item banks. This indicates that there is no need for establishing empirical thresholds through Monte Carlo simulations [51]. The McFadden's pseudo $R^2$ thresholds from the Monte Carlo simulations were all very small (≤0.0004 for anxiety and ≤0.0003 for depression), and as this is an effect size measure, applying a threshold that is substantially less than what would be considered a small but

**Table 1. Sociodemographic characteristics of participants and the Dutch general population.**

| Sociodemographic characteristic | Study participants* (n = 1002) | Dutch adult population 2013[a] (n = 13.3 million) |
|---|---|---|
| Age in years, mean ± SD (range) | 49 ± 17 (18–100) | |
| 18–39 | 34.3 | 34 |
| 40–64 | 44.4 | 44 |
| ≥65 | 21.3 | 22 |
| Gender | | |
| Male | 47.9 | 49 |
| Female | 52.1 | 51 |
| Educational level | | |
| Low | 32.0 | 32 |
| Middle | 39.9 | 40 |
| High | 28.0 | 28 |
| Region of residence | | |
| North | 11.5 | 10 |
| East | 20.5 | 21 |
| South | 21.5 | 22 |
| West | 46.6 | 47 |
| Ethnicity | | |
| Native | 79.6 | 80 |
| 1st and 2nd generation western immigrant | 12.6 | 10 |
| 1st and 2nd generation non-western immigrant | 7.8 | 10 |
| Living situation | | |
| Single | 29.2 | |
| Married/living together | 60.0 | |
| Relationship, not living together | 4.0 | |
| Living with parents | 5.7 | |
| Other | 1.1 | |
| Currently treated for psychological complaints | | |
| Yes | 10.1 | |
| No | 89.9 | |

* all results expressed as % unless otherwise noted.

SD: standard deviation;

[a] Based on data from statistics Netherlands (https://www.cbs.nl)

meaningful effect (e.g. 0.02) would not be meaningful according to any standard [51]. There-fore, the nominal α level of 0.01 and the McFadden's pseudo $R^2$ value of 0.02 were maintained. Table 2 shows the results of the DIF analyses. Two items in the anxiety item bank, 'It scared me when I felt nervous' (EDANX03) and 'I felt worried' (EDANX30), showed uniform and non-uniform DIF, respectively. The item 'I felt worried' is present in the PROMIS anxiety 7a short form. The items are present in respectively 1 and 3% of the CAT-based assessments. In the depression item bank, two items showed uniform DIF: 'I felt worthless' (EDDEP04) and 'I felt unhappy' (EDDEP36). Both these items are present in the PROMIS depression 6a, 8a and 8b short forms. The item 'I felt worthless' is also present in the PROMIS depression 4a short form. The item 'I felt worthless' is present in 8% of the CAT-based assessments, whereas the item 'I felt unhappy' is present in all CAT-based assessments. For the item 'It scared me when I felt nervous', the threshold parameters for the Dutch population were mostly slightly lower than the thresholds for the US population, indicating that the Dutch population endorses

**Table 2. McFadden's pseudo R² and IRT parameters for items displaying DIF.**

| Item bank | Item with DIF | DIF type | McFadden's pseudo R² | Slope; and threshold parameters | Included in CAT[d] |
|---|---|---|---|---|---|
| Anxiety | EDANX03: It scared me when I felt nervous | Uniform | $R^2_{12} = 0.021$ $R^2_{23} = 0.011$ | **NL**: 2.62; 0.15, 0.97, 2.02 | 1% |
| | | | | US: 3.74; 0.59, 1.18, 1.95 | |
| | EDANX30: I felt worried[a] | Non-uniform | $R^2_{12} = 0.010$ $R^2_{23} = 0.033$ | NL: 2.16; -1.12, -0.10, 1.29, 2.64 US: 3.14; -0.57, 0.24, 1.22, 2.12 | 3% |
| Depression | EDDEP04: I felt worthless[b] | Uniform | $R^2_{12} = 0.024$ $R^2_{23} = 0.013$ | **NL**: 2.93; -0.17, 0.58, 1.56, 2.61 US: 4.37; 0.29, 0.88, 1.61, 2.36 | 8% |
| | EDDEP36: I felt unhappy[c] | Uniform | $R^2_{12} = 0.037$ $R^2_{23} = 0.001$ | NL: 4.21; -0.14, 0.61, 1.33, 2.20 **US**: 3.44; -0.64, 0.23, 1.20, 2.17 | 100% |

The bold population had lower thresholds compared to the other population, indicating that this population endorses higher item response categories at the same level of the domain (anxiety, depression)

[a] present in the anxiety 7a short form

[b] present in the depression 4a, 6a, 8a and 8b short form

[c] present in the depression 6a, 8a and 8b short form

[d] Based on 4047 CAT-based assessments for anxiety and 4293 CAT-based assessments for depression

higher response categories at the same level of anxiety. The same applied for the item 'I felt worthless'. For the item 'I felt unhappy', the threshold parameters for the Dutch population were slightly higher than the thresholds for the US population, indicating that the Dutch population endorses lower response categories at the same level of depression. Fig 1 illustrates the impact of DIF on respondents total scores. The plots on the left show the impact of DIF when all items are considered, whereas the plots on the right show the impact of DIF when only DIF items are considered. The plots show that DIF had a minimal impact on the total score when all items are administered in each item bank. S1 Fig shows the impact of DIF on item scores per group for the items displaying DIF.

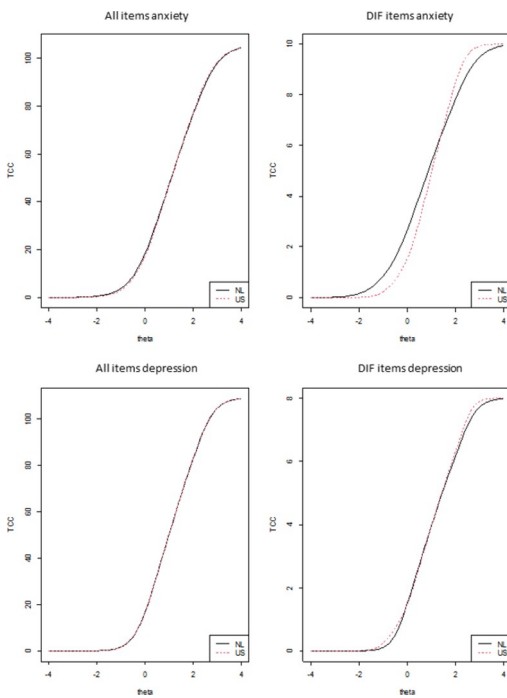

**Fig 1. Total impact of DIF on the test characteristic curve (TCC) for anxiety and depression.**

Table 3 displays the impact of DIF on T-scores of item banks, and short forms wherein DIF items are present. On a population level, mean anxiety T-scores based on hybrid parameters were approximately 0.5 point lower than T-scores based on the original US parameters, both for the full item bank as the short form. Differences on a population level were even smaller for depression T-scores of item banks and most short forms. Only for the 4a short form, mean depression T-scores based on the hybrid parameters were 1 point lower than T-scores based on the original US parameters. On an individual level, absolute T-score differences between the original and hybrid approach for anxiety ranged from 0 to 1.7 for the full item bank and from 0 to 1.9 for the short form. Absolute T-score differences between the original and hybrid approach for the depression item banks and most short forms ranged from 0 to 1.2 for individuals. A maximum T-score difference between the two approaches of 2.6 was found for the depression short form 4a. S2 Fig shows the difference for each full item bank and short form in relation to the T-score of individuals. Notably, the largest differences were found for participants with T-scores on the lower end of the scale.

Dutch reference values for anxiety and depression and comparisons with the US general population, using the original US item parameters, are presented in Table 4. Differences between T-scores of the Dutch general population and T-scores of the US general population for anxiety and depression were small (difference of 0.1 and 0.4 for anxiety and depression, respectively). Differences between T-scores of the Dutch general population and T-scores of the US general population for age-range and gender subpopulations were also small (differences between 0.1 and 0.7 for anxiety and between 0.1 and 1.4 for depression). T-scores of the Dutch general population and US general population showed similar patterns, with males scoring lower (i.e. less anxious or depressed) than females and lower scores for older age groups.

Using 0.5, 1 and 2 standard deviations, thresholds for mild, moderate and severe anxiety were set to 55, 60 and 70 respectively. The same thresholds applied for depression (Fig 2). When these thresholds were applied to anxiety T-scores of participants, 70% fell within normal limits (i.e. ≤55), 14% had mild symptoms (i.e. 56–60), 15% had moderate symptoms (i.e. 61–70) and 1% had severe symptoms (i.e. >70). For depression, 71% fell within normal limits (i.e. ≤55), 15% had mild symptoms (i.e. 56–60), 13% had moderate symptoms (i.e. 61–70) and 1% had severe symptoms (i.e. >70).

Table 3. PROMIS anxiety and depression T-scores[a] based on different sets of item parameters for different versions of the instruments.

| Version | Mean population T-score (SD) original approach[b] | Mean population T-score (SD) hybrid approach[c] | Mean absolute T-score difference (SD), [range] |
|---|---|---|---|
| Anxiety | | | |
| Full item bank | 49.9 (10.1) | 49.5 (10.3) | 0.40 (0.29) [0.00–1.67] |
| Short form 7a | 50.3 (9.2) | 49.8 (9.6) | 0.59 (0.48) [0.00–1.88] |
| Depression | | | |
| Full item bank | 49.6 (10.0) | 49.7 (9.9) | 0.15 (0.09) [0.00–0.73] |
| Short form 4a | 50.9 (8.5) | 49.9 (8.7) | 1.02 (0.52) [0.00–2.57] |
| Short form 6a | 49.7 (9.3) | 49.8 (9.0) | 0.52 (0.29) [0.00–1.24] |
| Short form 8a | 50.3 (9.2) | 50.4 (8.9) | 0.42 (0.25) [0.01–1.10] |
| Short form 8b | 50.2 (9.3) | 50.2 (9.1) | 0.38 (0.22) [0.00–1.15] |

SD: standard deviation

[a] T-scores, higher scores represent more anxiety/depression

[b] All items have the US item parameters

[c] Non-DIF items have the US item parameters, DIF items have the Dutch item parameters rescaled to the US metric

**Table 4. PROMIS anxiety and depression Dutch reference values[a] by age and gender and comparisons with the US general population [58].**

| | | Anxiety | | | Depression | | |
|---|---|---|---|---|---|---|---|
| | N Dutch population (%) | N US population (%) | Dutch mean T-score (SD) | US mean T-score (SD) | N US population (%) | Dutch mean T-score (SD) | US mean T-score (SD) |
| Total | 1002 (100) | 2724 (100) | 49.9 (10.1) | 50.0 (10.0) | 2160 (100) | 49.6 (10.0) | 50.0 (10.0) |
| Gender | | | | | | | |
| Male | 480 (48) | 1069 (39) | 49.0 (10.0) | 48.6 (9.5) | 890 (41) | 48.8 (10.1) | 48.7 (9.7) |
| Female | 522 (52) | 1654 (61) | 50.6 (10.1) | 50.9 (10.2) | 1269 (59) | 50.4 (9.9) | 50.9 (10.1) |
| Age in years | | | | | | | |
| 18–34 | 253 (25) | 659 (24) | 51.8 (9.9) | 52.4 (10.7) | 496 (23) | 52.0 (9.3) | 52.3 (10.9) |
| 35–44 | 147 (15) | 496 (18) | 51.4 (10.9) | 50.9 (11.1) | 366 (17) | 50.5 (10.8) | 50.6 (10.9) |
| 45–54 | 173 (17) | 417 (15) | 50.0 (10.9) | 50.1 (9.5) | 359 (17) | 50.0 (11.0) | 50.8 (10.0) |
| 55–64 | 216 (22) | 442 (16) | 48.9 (9.4) | 49.3 (9.5) | 373 (17) | 48.8 (9.8) | 49.5 (9.7) |
| 65–74 | 191 (19) | 365 (13) | 47.5 (9.1) | 48.1 (8.8) | 290 (13) | 47.0 (8.9) | 48.4 (8.8) |
| 75+ | 22 (2) | 345 (13) | 46.2 (9.4) | 46.9 (7.9) | 276 (13) | 46.0 (9.6) | 46.5 (7.2) |

SD: standard deviation

[a] T-scores, higher scores represent more anxiety/depression; T-scores were calculated based on the original US item parameters

## Discussion

This study assessed DIF for language between the Netherlands and the US for the PROMIS Anxiety and Depression item banks, and presented Dutch reference values for the general population and relevant subpopulations. We found some items with DIF, but the impact of DIF on population level T-scores was small, both for full item banks as for short forms. This supports the applicability of the US scoring algorithm in the Netherlands and strengthens the cross-cultural validity of the Dutch-Flemish PROMIS Anxiety and Depression item banks. It enables the comparison of scores between Dutch and US populations. The established Dutch reference values can be used to interpret symptoms of anxiety and depression in research and clinical practice.

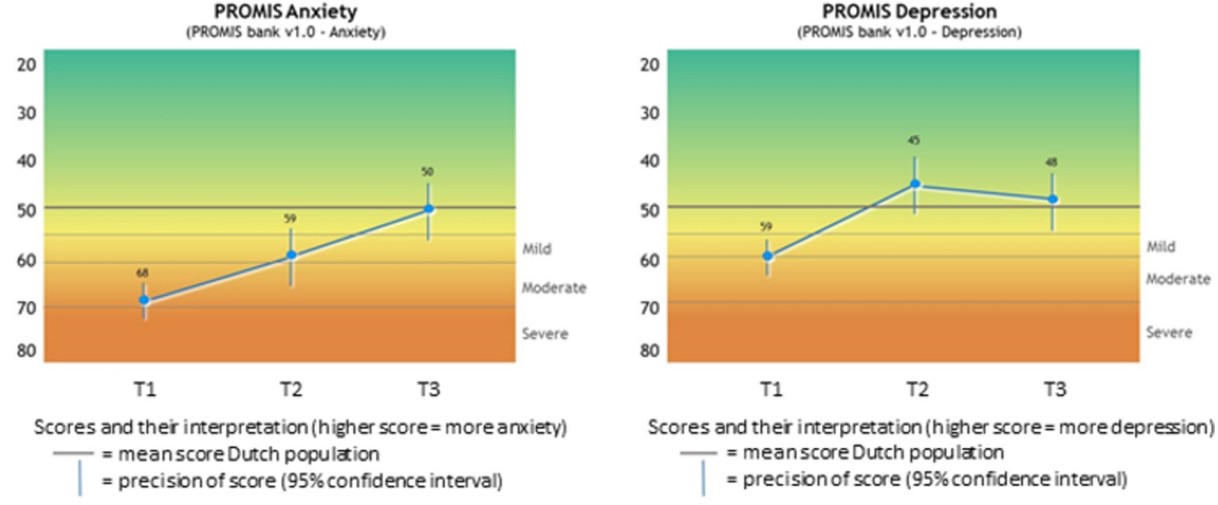

**Fig 2. Visual feedback [59] of PROMIS Anxiety and Depression scores, based on Dutch mean T-scores and Dutch thresholds for mild, moderate and severe symptoms.** The blue lines represent imaginary data showing the course of symptoms over three consecutive assessments (T1, T2 and T3).

We only found a limited number of items with DIF, which had negligible impact on total scores when all items in the item banks were administered. However, the impact of DIF might be more in short forms wherein DIF items were present, because only a small number of items are administered. On a population level, the impact of DIF on T-scores was small for all short forms. Because DIF in the two depression items had opposite direction, the effect of DIF might have been canceled out in the short forms in which both items were present. On individual level the impact was larger, especially for the depression short form 4a, with a maximum difference of 2.6 points. This is close to the amount of 3 points that is generally considered minimally important [60–63], and therefore this short form might not be the best option to assess symptoms of depression in individuals. Most DIF items were not frequently administered in CAT-based assessments, but the item 'I felt unhappy' was present in all CATs [32]. In a future study, it might be interesting to explore the impact of DIF on CAT T-scores and also to explore whether omitting DIF items from the item bank could result in equally precise scores and similar amount of items administered.

DIF for language could be caused by a lack of translational equivalence [64]. In the development of PROMIS measures generally a translatability review is performed, but only for Spanish. In a translatability review, the original measure is reviewed to determine its suitability for future translations. A translatability review is best conducted as early as possible during the development of a new measure, preferably before quantitative testing, as changes to the measure can still be made at this point. To make new PROMIS measures more applicable for translations to other languages, which is increasingly occurring, a broader translatability review might be a useful additional step in the developmental process of PROMIS measures. During the Dutch-Flemish translation process of the PROMIS Anxiety and Depression item banks, no particular difficulties were experienced with translating the items showing DIF for language [42]. The items 'I felt worried' and 'I felt unhappy' also showed DIF for language in a study comparing the Brazilian to the US version [41], but studies in Germany and Spain found different DIF items [65, 66]. Although a translatability review might reduce translation difficulties, it does not replace the evaluation of DIF for language, as DIF can also occur due to cultural differences [67]. Therefore DIF studies are recommended after every translation [52].

The negligible impact of DIF made it possible to compare item bank scores of the Dutch general population to the US general population. T-scores of the Dutch general population were similar to scores of the US general population, both for the total population (difference of 0.1 for anxiety and 0.4 for depression) and for age-range and gender subpopulations. Unfortunately, it is not clear yet what a minimal important difference is in scores between groups for anxiety and depression [68], although most studies suggest a within-person change of at least 3 points to be meaningful [60–63]. Thus, we think it is safe to conclude that T-scores of the Dutch general population were similar to scores of the US general population. Because of similarity in scores and standard deviations, thresholds for mild, moderate and severe symptoms of anxiety and depression were identical to the thresholds based on the US data [57, 63].

The inclusion of anxiety and depression outcomes in many ICHOM Standard Sets shows that measuring anxiety and depression is relevant for many patient groups and persons without diseases, and not only those with mental disorders [6, 7]. In the Standard Set for Overall Adult Health, it is advocated to measure anxiety and depression via the PROMIS Scale v1.2 – Global Health [69], resulting in a global mental health score [70], for which Dutch reference values recently have been published [48]. In the other 16 Standard Sets that include anxiety and depression, a range of PROMs is advocated, including disease-specific PROMs, cancer-specific PROMs, anxiety/depression-specific PROMs and generic PROMs [6, 7]. A more universal and standardized approach to measuring anxiety and depression will facilitate outcome measurement in clinical practice and comparisons of scores across patient groups [14, 15].

PROMIS anxiety and depression instruments offer opportunities here, and the results of this study expands their utility.

PROMIS anxiety and depression instruments have several advantages over current legacy instruments for anxiety and depression. First, PROMIS instruments are applicable across the general population and various patient groups, as well as those patients with multimorbidity, rare diseases, or without a definite diagnoses [17, 18, 20]. This enables the comparison of patient groups, benchmarking and improving the quality of care. Second, PROMIS Anxiety and Depression item banks can be used as CAT, which reduces the response burden while high measurement precision is maintained, and as such is valuable in clinical practice [23]. Currently a limited number of countries, including the Netherlands, have access to technical solutions for CAT applications, but this is expected to expand rapidly in the near future. Third, several crosswalk studies have linked scores of legacy instruments to PROMIS anxiety and depression instruments [71–75], which facilitates the uptake of PROMIS instruments and the interpretation of scores, even when legacy instruments have been used in the past. Last, PROMIS is a sustainable state-of-the-art measurement system that is actively maintained by the PROMIS Health Organization, in order to facilitate the widespread use and adoption of PROMIS in research and clinical practice.

A strength of the present study is that we not only assessed the impact of DIF when all items were considered, but applied Stocking-Lord constants to investigate the impact of DIF on T-scores of full item banks and short forms. Moreover, the large sample size made sure that the Dutch reference values have been estimated reliably. However, some subgroups (especially adults ages 75 years and older) were relatively small, which can be considered a limitation. Second, although our sample was broadly representative for the Dutch general population on some important characteristics, we cannot be certain that this is also the case for other important characteristics, such as income level and employment status. One could argue that persons who have the time to participate in an internet panel and complete item banks, might more often be persons without full-time employment, which might in turn be caused by physical or mental problems. The non-probabilistic selection procedure might have had an impact on the general population reference scores presented in this article.

## Conclusions

The limited number of items with DIF in PROMIS Anxiety and Depression item banks, having small impact on population T-scores, supports the applicability of the US scoring algorithm and enables the comparison of scores of the Dutch and US population. The Dutch general population had a T-score of 49.9 for anxiety and 49.6 for depression, similar to the T-scores of 50 of the US general population. The Dutch reference values reported in this study provide an important tool for healthcare professionals and researchers to evaluate and interpret symptoms of anxiety and depression. The presented reference values for subpopulations allow a more tailored and relevant interpretation and understanding of symptoms of anxiety and depression. Incorporating the Dutch reference values and thresholds in the feedback patients and healthcare professionals receive regarding their mental health status as assessed with PROMIS anxiety and depression instruments, will facilitate interpretation of scores by patients and healthcare professionals. The availability of Dutch reference values may stimulate the uptake of PROMIS instruments for anxiety and depression, and contribute to standardized measurements of anxiety and depression.

## Supporting information

**S1 Fig. Category response curves for items displaying DIF.**
(TIF)

**S2 Fig. Relation between T-scores and differences in T-scores original vs. hybrid approach.**
(TIF)

**S1 Data. Anxiety data of respondents.**
(POR)

**S2 Data. Depression data of respondents.**
(POR)

## Author Contributions

**Conceptualization:** Edwin de Beurs, Leo D. Roorda, Caroline B. Terwee.

**Data curation:** Gerard Flens.

**Formal analysis:** Ellen B. M. Elsman.

**Investigation:** Ellen B. M. Elsman, Gerard Flens, Caroline B. Terwee.

**Methodology:** Caroline B. Terwee.

**Resources:** Caroline B. Terwee.

**Supervision:** Caroline B. Terwee.

**Validation:** Gerard Flens.

**Visualization:** Ellen B. M. Elsman.

**Writing – original draft:** Ellen B. M. Elsman.

**Writing – review & editing:** Gerard Flens, Edwin de Beurs, Leo D. Roorda, Caroline B. Terwee.

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
