## [Decision Letter · Decision Letter 0]

16 Dec 2021

PONE-D-21-18689Towards standardization of measuring anxiety and depression: Differential item functioning for language and Dutch reference values of PROMIS item banksPLOS ONE

Dear Dr. Terwee,

Thank you for submitting your manuscript to PLOS ONE. After careful consideration, we feel that it has merit but does not fully meet PLOS ONE’s publication criteria as it currently stands. Therefore, we invite you to submit a revised version of the manuscript that addresses the points raised during the review process. The manuscript has been evaluated by two reviewers, and their comments are available below.The reviewers have raised a number of concerns that need attention, and they request additional information on methodological aspects of the study and analyses, as well as additional discussion and contextaualization of the current work. Could you please revise the manuscript to carefully address the concerns raised? Please submit your revised manuscript by Jan 28 2022 11:59PM. If you will need more time than this to complete your revisions, please reply to this message or contact the journal office at plosone@plos.org. Please include the following items when submitting your revised manuscript:

We look forward to receiving your revised manuscript.

Kind regards,

Vanessa Carels

Staff Editor

PLOS ONE

Journal Requirements:

2. Thank you for stating the following in the Competing Interests/Financial Disclosure section:

“CB Terwee and LD Roorda are members of the PROMIS Health Organization and the Dutch-Flemish PROMIS National Center, which aim to improve health outcomes by developing, maintaining, improving, and encouraging the application of PROMIS in research and clinical practice. The other authors have no conflict of interest.”

We note that one or more of the authors are employed by a commercial company: PROMIS Health Organization and the Dutch-Flemish PROMIS National Center

Reviewers' comments:

Reviewer's Responses to Questions

**Comments to the Author**

1. Is the manuscript technically sound, and do the data support the conclusions?

Reviewer #1: Yes

Reviewer #2: Yes

2. Has the statistical analysis been performed appropriately and rigorously? 

Reviewer #1: Yes

Reviewer #2: Yes

3. Have the authors made all data underlying the findings in their manuscript fully available?

Reviewer #1: Yes

Reviewer #2: Yes

4. Is the manuscript presented in an intelligible fashion and written in standard English?

Reviewer #1: Yes

Reviewer #2: Yes

5. Review Comments to the Author

Reviewer #1: Thank you for the opportunity to review your manuscript on testing DIF by language for the PROMIS Depression and Anxiety measures. Overall, the paper is well-written and the methods are rigorous. There were a few places that could use a bit more clarity and expanded discussion (noted below) but overall the manuscript provides a nice addition to the PROM literature on the use of PROMIS instruments across different countries.

1. As someone who has familiarity with developing and using PROMIS instruments, I had an easy time following along but for readers who do not have such familiarity, you might want to explicitly point them to some of the early PROMIS papers that describe the initiative instead of just citing them (current references 17 and 18 could be called out in the text with something such as, "see [17, 18] for an overview of PROMIS for additional insight into the aims and early findings of this initiative" or something like that.

2. Even with my familiarity, there were a few places where I got confused. I went into this assuming there were calibrated and normed item parameters for the Dutch version of the anxiety and depression instruments but now I'm not sure there actually are. Maybe this could be stated upfront? I got confused in the results section (pg. 12, line 254) as to whether you were talking about the Dutch sample scored on the US metric or the Dutch sample scored on the Dutch metric (if there is one?). I think the nuanced difference is that you refer to the Dutch general population versus reference population, but you do say “reference values for…the Dutch general population” so that muddies things. I think this part is referring to the Dutch sample scored with the US algorithm compared to the US reference scores but this could be clarified, especially for people who are not familiar with the PROMIS terminology. Even just saying it bluntly somewhere in the results, e.g., "T-scores between the Dutch sample scored with the US algorithm compared to the US reference T-scores…"

3. In general, the discussion doesn’t really focus on the applicability of the findings and mostly restates the introduction. It would be helpful to focus more on how performing this DIF analysis relates to being able to use the PROMIS measures - does it expand their utility to enable reliable and valid global comparison between two countries? Does it say anything about language differences across the Netherlands and the US that may be important for consideration when developing new measures or interpreting current ones? Generally we do a Spanish translatability review so it’s unsurprising that the Spain DIF didn’t find differences - do your results suggest the US development process should include a broader translatability review? Or are there too many natural linguistic differences across countries that such an initiative would be too hard to conduct and better to consider via DIF afterwards? For the Netherlands specifically, did any of the US items have to be translated to have close but not the exact same wording that might be playing a role here? I reviewed the Terwee et al. 2014 paper and it doesn't look like there were issues for Anxiety and Depression but may be worth referring back to in this paper as a reason to retain the items with minimal DIF as well. Overall, I think the discussion could be strengthened by including more considerations like these instead of just restating what's in the introduction.

Reviewer #2: This is a methodological article on the performance of the Dutch version of two PROMIS domains on the mental dimension of health, the PROMIS depression and Anxiety domains. The article has two main objectives: firstly, it aims to assess differential item functioning of the Dutch PROMIS Depression and Anxiety item banks compared to the original English version; and second, to provide population reference norms. The article is relevant as it provides further evidence on the validity and comparability of the Dutch version of these measures for their use in an international context. Additionally, population reference norms obtained for the Dutch population provide additional aids to facilitate interpretation of the scores within the Dutch context.

The article is clearly written and follows state of the art methods to pursue it objectives, and a large, adequately sized sample is used. Here I include some comments intended to provide the authors with suggestions to enhance understanding of the methods followed in some sections and improve the overall quality of the manuscript.

Specific comments:

- Participants and Methods, page 6: The sample was selected from an internet panel. Although evidence is presented that the sample has broadly the same distribution as the Dutch adult population with regard to main variables (age, sex, education level, region and ethnicity), my main concern has to do with the representativeness of the sample as it was obtained using non-probabilistic methods where not all individuals on the Dutch general population had equal probability of selection. This selection procedure may have an impact on the general population Norms presented, and should at least be acknowledged in the limitations section of the article.

- Statistical analysis, page 7: A reference to justify the selected cut-off point used to determine DIF on the McFadden’s pseudo R2 should be added. Moreover, more details on the Monte Carlo simulation, its usefulness and the results obtained regarding these simulations should be provided. Are the Monte Carlo simulations applied those implemented in the lordif package in R? If so, these simulations generate empirical distributions of the McFadden’s pseudo R2 statistic under no-DIF conditions and preserving observed group differences in the ability level. This would serve to determine an empirical cut-off point for the statistic used to determine DIF. How many replications were conducted? Which were the results of the Monte carlo simulations? How did the Monte Carlo simulations results differ from the threshold used for the McFadden’s pseudo R2?

- Statistical analysis, page 8: the part describing equating of the Dutch item parameters with DIF to US metric using Stocking and Lord transformation is disproportionately long, and I would recommend to reduce it.

- Statistical analysis, page 8: “T-scores were calculated with the original US item parameters, as well as with a hybrid set of item parameters, and subsequently compared”: it should be indicated how the T-scores were obtained, i.e. was Expected a posteriori (EAP) estimation from the GRM model used? .

- Statistical analysis, page 8: “To investigate the impact of DIF on CATs, it was assessed how often the DIF items were included in CATs, based on 4047 CATs for anxiety and 4293 CATs for depression from an ongoing study [32].”. Please, clarify the source of the sample from this study. Is it general population or a patients’ sample? The items selected for a CAT may depend on the ability of the individual assessed.

- Discussion, page 15, second paragraph: “The inclusion of anxiety and depression outcomes in many ICHOM standard Sets shows that measuring anxiety and depression is relevant for many patient groups, and not only those with mental disorders (…). A more universal and standardized approach to measuring anxiety and depression will facilitate outcome measurement in clinical practice and comparisons of scores across patient groups [14, 15]. PROMIS anxiety and depression instruments offer opportunities here.”. This paragraph is quite redundant with the first paragraph of the introduction and it does not add additional information to the discussion related to the results obtained. Therefore, I suggest to eliminate it or reduce it.

Minor comments:

- Participants and procedures, page 6: “Participants needed to be representative for the Dutch general population with respect to age distribution, gender, educational level (low, middle, high), region of residence (north, east, south, west) and ethnicity (native Dutch, first- and second-generation western immigrant, first- and second generation non-western immigrant).

- Measures section, page 6, description of the PROMIS Item Banks V1.0 - Anxiety and Depression: The reference for the article where the development of the PROMIS mental health domains (Pilkonis et al,2021. doi:10.1177/1073191111411667) should be added here.

6. PLOS authors have the option to publish the peer review history of their article (what does this mean?). If published, this will include your full peer review and any attached files.

Reviewer #1: No

Reviewer #2: No

---

## [Author Response · Author response to Decision Letter 0]

10 Jan 2022

Please note that we have also uploaded a 'response to reviewers' Word document.

PLOS ONE

Staff editor: Vanessa Carels

10 January 2022

Manuscript ID: PONE-D-21-18689

Title: Towards standardization of measuring anxiety and depression: Differential item functioning for language and Dutch reference values of PROMIS item banks

Dear dr. Carels, 

First of all, we would like to thank you for the opportunity to submit a revised version of our manuscript. We thank the reviewers for their careful examination of the manuscript and useful suggestions to improve the quality of the paper. Below is our point-by-point response to each of the reviewers’ comments. Our manuscript contains tracked changes to highlight all changes that have been made to the manuscript. Additionally, we have included an unmarked version of the manuscript without tracked changes. Please note that the page numbers in the authors’ reply refer to the revised manuscript with tracked changes.

Journal Requirements:

We have taken careful consideration that our revised manuscript meets the PLOS ONE style requirements.

2. Thank you for stating the following in the Competing Interests/Financial Disclosure section:

“CB Terwee and LD Roorda are members of the PROMIS Health Organization and the Dutch-Flemish PROMIS National Center, which aim to improve health outcomes by developing, maintaining, improving, and encouraging the application of PROMIS in research and clinical practice. The other authors have no conflict of interest.”

We note that one or more of the authors are employed by a commercial company: PROMIS Health Organization and the Dutch-Flemish PROMIS National Center

We have updated the Author Contributions section in the online submission form.

Thank you for changing this information on our behalf. For the Funding statement, the following text can be inserted:

“The PROMIS Health Organization is a non-profit charitable foundation and the Dutch-Flemish PROMIS National Center is a network of local members of the PHO who are developing or applying PROMIS measures in the Netherlands or Belgium. Both organizations did not provide support in the form of salaries for authors CT and LR, and did not have any additional role in the study design, data collection and analysis, decision to publish, or preparation of the manuscript. The specific roles of these authors are articulated in the ‘author contributions’ section.”

For the Competing Interest statement, the following text can be inserted:

“CB Terwee and LD Roorda are members of the PROMIS Health Organization and the Dutch-Flemish PROMIS National Center, which aim to improve health outcomes by developing, maintaining, improving, and encouraging the application of PROMIS in research and clinical practice. This does not alter our adherence to PLOS ONE policies on sharing data and materials. The other authors have no conflict of interest.”

We have linked the corresponding author’s ORCID ID to editorial manager.

We have reviewed our reference list and updated the references that were previously indicated as ‘submitted’ but at this time have been published. We could not identify any papers that have been retracted in our reference list.

Review Comments to the Author

Reviewer #1: 

Thank you for the opportunity to review your manuscript on testing DIF by language for the PROMIS Depression and Anxiety measures. Overall, the paper is well-written and the methods are rigorous. There were a few places that could use a bit more clarity and expanded discussion (noted below) but overall the manuscript provides a nice addition to the PROM literature on the use of PROMIS instruments across different countries.

We thank the reviewer for the compliments and will reply to the more detailed comments below.

1. As someone who has familiarity with developing and using PROMIS instruments, I had an easy time following along but for readers who do not have such familiarity, you might want to explicitly point them to some of the early PROMIS papers that describe the initiative instead of just citing them (current references 17 and 18 could be called out in the text with something such as, "see [17, 18] for an overview of PROMIS for additional insight into the aims and early findings of this initiative" or something like that.

We agree with the reviewer that less familiar readers might benefit from explicitly pointing them towards tome of the early PROMIS papers, and have changed the manuscript accordingly (page 3-4).

2. Even with my familiarity, there were a few places where I got confused. I went into this assuming there were calibrated and normed item parameters for the Dutch version of the anxiety and depression instruments but now I'm not sure there actually are. Maybe this could be stated upfront? I got confused in the results section (pg. 12, line 254) as to whether you were talking about the Dutch sample scored on the US metric or the Dutch sample scored on the Dutch metric (if there is one?). I think the nuanced difference is that you refer to the Dutch general population versus reference population, but you do say “reference values for…the Dutch general population” so that muddies things. I think this part is referring to the Dutch sample scored with the US algorithm compared to the US reference scores but this could be clarified, especially for people who are not familiar with the PROMIS terminology. Even just saying it bluntly somewhere in the results, e.g., "T-scores between the Dutch sample scored with the US algorithm compared to the US reference T-scores…"

We thank the reviewer for this valuable comment. Although Dutch item parameters were used in the Netherlands for a while as an experiment, they are no longer used. The T-scores of the anxiety and depression instruments were therefore calculated in this study using the US calibrated and normed item parameters, which is standard practice for PROMIS measures (see also highlighted sentence page 7). This is also the reason we wanted to investigate DIF for language, because only if items are free of DIF, the US scoring algorithm is appropriate to use and scores between countries can be compared. We estimated Dutch item parameters from a graded response model fitted on the Dutch general population sample (see also highlighted sentence page 8), and have used these to obtain a hybrid set of parameters for those items that had DIF for language. We could then compare the scores resulting from the hybrid set of item parameters with the scores resulting from the original US item parameters. For the comparison of scores between the Dutch population and the US population, we have calculated T-scores with the original US item parameters (see also highlighted sentence on page 9). We agree with the reviewer that some information could be stated more upfront and can be clarified in relevant parts throughout the manuscript. We also agree that providing reference values for the Dutch population and comparing these to the US reference population is confusing. We have changed the manuscript to resolve these issues (page 2, 5, 7, 8, 9, 13, 15, 16, 18).

3. In general, the discussion doesn’t really focus on the applicability of the findings and mostly restates the introduction. It would be helpful to focus more on how performing this DIF analysis relates to being able to use the PROMIS measures - does it expand their utility to enable reliable and valid global comparison between two countries? Does it say anything about language differences across the Netherlands and the US that may be important for consideration when developing new measures or interpreting current ones? Generally we do a Spanish translatability review so it’s unsurprising that the Spain DIF didn’t find differences - do your results suggest the US development process should include a broader translatability review? Or are there too many natural linguistic differences across countries that such an initiative would be too hard to conduct and better to consider via DIF afterwards? For the Netherlands specifically, did any of the US items have to be translated to have close but not the exact same wording that might be playing a role here? I reviewed the Terwee et al. 2014 paper and it doesn't look like there were issues for Anxiety and Depression but may be worth referring back to in this paper as a reason to retain the items with minimal DIF as well. Overall, I think the discussion could be strengthened by including more considerations like these instead of just restating what's in the introduction.

The reviewer makes some valuable suggestions that will strengthen our discussion section. With respect to the utility of PROMIS measures and comparing scores between two countries, we believe that the DIF analyses and the minimal impact on population level T-scores supports the applicability of the US scoring algorithm and strengthens the validity of the Dutch-Flemish item banks. Because of the results of the DIF analyses, we were able to validly compare Dutch population’ scores with US population’ scores. We have adapted the first paragraph of the discussion to reflect on this (page 14). Conducting a broader translatability review for new PROMIS measures than just Spanish might be a valuable additional step in the developmental process, especially because PROMIS measures are increasingly being translated to many different languages. However, although it might reduce difficulties with translations of items, it does not replace the evaluation of DIF for language after a measure has been developed and translated because DIF may also occur because of cultural differences . For the DIF items in our study, no particular difficulties were encountered during their translation, but it is remarkable that some of these items also showed DIF for language in other countries. We have elaborated on this in the discussion (page 15). We have also adapted the introduction and discussion to diminish the amount of overlap in these sections (page 3, 14-15, 16).

Reviewer #2: 

This is a methodological article on the performance of the Dutch version of two PROMIS domains on the mental dimension of health, the PROMIS depression and Anxiety domains. The article has two main objectives: firstly, it aims to assess differential item functioning of the Dutch PROMIS Depression and Anxiety item banks compared to the original English version; and second, to provide population reference norms. The article is relevant as it provides further evidence on the validity and comparability of the Dutch version of these measures for their use in an international context. Additionally, population reference norms obtained for the Dutch population provide additional aids to facilitate interpretation of the scores within the Dutch context. The article is clearly written and follows state of the art methods to pursue it objectives, and a large, adequately sized sample is used. Here I include some comments intended to provide the authors with suggestions to enhance understanding of the methods followed in some sections and improve the overall quality of the manuscript.

We thank the reviewer for acknowledging the relevance of our article and the other compliments. We reply to the more detailed comments below. 

Specific comments:

- Participants and Methods, page 6: The sample was selected from an internet panel. Although evidence is presented that the sample has broadly the same distribution as the Dutch adult population with regard to main variables (age, sex, education level, region and ethnicity), my main concern has to do with the representativeness of the sample as it was obtained using non-probabilistic methods where not all individuals on the Dutch general population had equal probability of selection. This selection procedure may have an impact on the general population Norms presented, and should at least be acknowledged in the limitations section of the article.

The reviewer makes a valid remark. Indeed, although the sample is broadly representative for the Dutch population on important characteristics, we cannot be sure this is true for other important characteristics, such as income levels, employment and presence of disease. For example, those who have time to participate in an internet panel and complete these item banks voluntarily, might more often be persons without full-time employment, for example caused by physical or mental complaints. We have now explained this in the discussion (page 17).

- Statistical analysis, page 7: A reference to justify the selected cut-off point used to determine DIF on the McFadden’s pseudo R2 should be added. Moreover, more details on the Monte Carlo simulation, its usefulness and the results obtained regarding these simulations should be provided. Are the Monte Carlo simulations applied those implemented in the lordif package in R? If so, these simulations generate empirical distributions of the McFadden’s pseudo R2 statistic under no-DIF conditions and preserving observed group differences in the ability level. This would serve to determine an empirical cut-off point for the statistic used to determine DIF. How many replications were conducted? Which were the results of the Monte carlo simulations? How did the Monte Carlo simulations results differ from the threshold used for the McFadden’s pseudo R2?

We thank the reviewer for these questions. We have added references to justify the selected cut-off point for the McFadden’s pseudo R2 (page 8). Indeed, the Monte Carlo simulations were those implemented in the lordif package in R. The Monte Carlo simulations implemented in the lordif package are driven by type I error control (i.e. control of false positive results). We conducted 1000 replications (this took approximately 2 hours per item bank on a new, high-speed computer), and found that the empirical threshold values for probability associated with the χ2 statistic were all close to the nominal α (=0.01) level: for anxiety mean 1-2=0.009, mean 1-3=0.01, mean 2-3=0.009, and for depression mean 1-2=0.01, mean 1-3=0.01, mean 2-3=0.01. This suggests that the type I error rate is well controlled, and there is no need for establishing empirical thresholds through Monte Carlo simulations . The resulting McFadden’s pseudo R2 thresholds from the Monte Carlo simulations were all very small (≤0.0004 for anxiety and ≤0.0003 for depression), and as this is an effect size measure, applying a threshold that is substantially less than what would be considered a small but meaningful effect (e.g. 0.02) would not be meaningful according to any standard3. Based on these results, we concluded that type I error rate was well controlled for both items banks and we maintained the nominal α level of 0.01 and the McFadden’s pseudo R2 value of 0.02. We have elaborated on this in the manuscript (page 8 and 10-11). 

- Statistical analysis, page 8: the part describing equating of the Dutch item parameters with DIF to US metric using Stocking and Lord transformation is disproportionately long, and I would recommend to reduce it.

We agree with the reviewer that this section could be reduced, and have done so accordingly (page 8-9). 

- Statistical analysis, page 8: “T-scores were calculated with the original US item parameters, as well as with a hybrid set of item parameters, and subsequently compared”: it should be indicated how the T-scores were obtained, i.e. was Expected a posteriori (EAP) estimation from the GRM model used? 

Indeed, expected a posteriori estimates from the GRM were used (see also highlighted sentence page 7). We have now added this to the above sentence as well (page 8). 

- Statistical analysis, page 8: “To investigate the impact of DIF on CATs, it was assessed how often the DIF items were included in CATs, based on 4047 CATs for anxiety and 4293 CATs for depression from an ongoing study [32].”. Please, clarify the source of the sample from this study. Is it general population or a patients’ sample? The items selected for a CAT may depend on the ability of the individual assessed.

It concerned a clinical population sample of adult patients who started outpatient treatment for common mental disorders. We have added this to the manuscript (page 6).

- Discussion, page 15, second paragraph: “The inclusion of anxiety and depression outcomes in many ICHOM standard Sets shows that measuring anxiety and depression is relevant for many patient groups, and not only those with mental disorders (…). A more universal and standardized approach to measuring anxiety and depression will facilitate outcome measurement in clinical practice and comparisons of scores across patient groups [14, 15]. PROMIS anxiety and depression instruments offer opportunities here.”. This paragraph is quite redundant with the first paragraph of the introduction and it does not add additional information to the discussion related to the results obtained. Therefore, I suggest to eliminate it or reduce it.

We agree with the reviewer that some of the information was redundant, and we have adapted the introduction and reduced this paragraph in the discussion to eliminate redundant information (page 3, page 16). 

Minor comments:

- Participants and procedures, page 6: “Participants needed to be representative for the Dutch general population with respect to age distribution, gender, educational level (low, middle, high), region of residence (north, east, south, west) and ethnicity (native Dutch, first- and second-generation western immigrant, first- and second generation non-western immigrant).

We are not sure what the reviewer means with this comment.

- Measures section, page 6, description of the PROMIS Item Banks V1.0 - Anxiety and Depression: The reference for the article where the development of the PROMIS mental health domains (Pilkonis et al,2021. doi:10.1177/1073191111411667) should be added here.

We have added this reference (page 6-7).

We hope we adequately addressed all points and comments made by the reviewers. Thank you for reconsidering our manuscript. We look forward to receiving your response in due course.

Yours sincerely,

The authors

---

## [Decision Letter · Decision Letter 1]

30 Mar 2022

PONE-D-21-18689R1Towards standardization of measuring anxiety and depression: Differential item functioning for language and Dutch reference values of PROMIS item banksPLOS ONE

Dear Dr. Terwee,

Thank you for submitting your manuscript to PLOS ONE. After careful consideration, we feel that it has merit but does not fully meet PLOS ONE’s publication criteria as it currently stands. Therefore, we invite you to submit a revised version of the manuscript that addresses the points raised during the review process.

Specifically, please address the remaining concerns from Reviewer 2.

We look forward to receiving your revised manuscript.

Kind regards,

Jianhong Zhou

Associate Editor

PLOS ONE

Journal Requirements:

Reviewers' comments:

Reviewer's Responses to Questions

**Comments to the Author**

1. If the authors have adequately addressed your comments raised in a previous round of review and you feel that this manuscript is now acceptable for publication, you may indicate that here to bypass the “Comments to the Author” section, enter your conflict of interest statement in the “Confidential to Editor” section, and submit your "Accept" recommendation.

Reviewer #1: All comments have been addressed

Reviewer #2: (No Response)

2. Is the manuscript technically sound, and do the data support the conclusions?

Reviewer #1: Yes

Reviewer #2: Yes

3. Has the statistical analysis been performed appropriately and rigorously? 

Reviewer #1: Yes

Reviewer #2: Yes

4. Have the authors made all data underlying the findings in their manuscript fully available?

Reviewer #1: Yes

Reviewer #2: Yes

5. Is the manuscript presented in an intelligible fashion and written in standard English?

Reviewer #1: Yes

Reviewer #2: Yes

6. Review Comments to the Author

Reviewer #1: Thank you for addressing all of my comments. The manuscript is a great contribution to the measurement science literature as well as to researchers and clinicians implementing PROMIS.

Reviewer #2: I really appreciate the author’s careful consideration of the reviewers’ comments and clear responses to them. I only have a few additional minor suggestions for further clarification (the indication of page and line numbers refer to the version with track changes):

- Page 5, line 107; and Page9, line 194: The indication that the reference values provided refer to the Dutch general population has been deleted. However, I think it is important to indicate which is the reference population for which the reference values are provided, as reference values (or sub-norms) can be provided for any relevant population (see norms and sub-norms chapter in PROMIS Score and Interpret section of the PROMIS web page : https://www.healthmeasures.net/score-and-interpret/interpret-scores/promis/reference-populations). Therefore, I would keep the indication that reference values obtained are Dutch general population-based reference values for the overall population and by age and gender.

- Page 7, line 143: “expressed as T-scores, with a mean of 50 and a standard deviation of 10 for the US general population [16]”: Please check this reference, as I think it does not refer to scoring interpretation of PROMIS. Should this be reference 17 instead?

- Pages 10 (line 216) and 11 (line 220), reference 58: Given this reference is a personal e-mail sent to the author, I don’t think this can be included among the references of the article.

- Page 12, line 248: “The Stocking-Lord constants were Α=0.7604 and Β=-0.1070 for anxiety and Α=0.7786 and Β=0.0339 for depression”: Following my suggestions the information regarding stocking and lord transformation has been substantially reduced, which I think is Good. Now, I don’t think it is relevant to include the SL constants A and B for anxiety and depression. Otherwise, if included, additional indication should be provided to indicate what A and B mean (perhaps keep the formulas)

7. PLOS authors have the option to publish the peer review history of their article (what does this mean?). If published, this will include your full peer review and any attached files.

Reviewer #1: No

Reviewer #2: **Yes: **Gemma Vilagut

---

## [Author Response · Author response to Decision Letter 1]

25 Apr 2022

Reviewer #1: 

Thank you for addressing all of my comments. The manuscript is a great contribution to the measurement science literature as well as to researchers and clinicians implementing PROMIS.

We are pleased to read that we have addressed the reviewer’s comments adequately and for the reviewer’s acknowledgement of the value of our manuscript.

Reviewer #2: 

I really appreciate the author’s careful consideration of the reviewers’ comments and clear responses to them. I only have a few additional minor suggestions for further clarification (the indication of page and line numbers refer to the version with track changes):

We thank the reviewer for these compliments and will respond to the additional suggestions below.

- Page 5, line 107; and Page9, line 194: The indication that the reference values provided refer to the Dutch general population has been deleted. However, I think it is important to indicate which is the reference population for which the reference values are provided, as reference values (or sub-norms) can be provided for any relevant population (see norms and sub-norms chapter in PROMIS Score and Interpret section of the PROMIS web page : https://www.healthmeasures.net/score-and-interpret/interpret-scores/promis/reference-populations). Therefore, I would keep the indication that reference values obtained are Dutch general population-based reference values for the overall population and by age and gender.

We thank the reviewer for this remark, and agree that the reference values are established for the Dutch general population and that this notion is important. The reason we changed this is because these reference values, although established for the Dutch general population, are often used in clinical care, to compare scores from clinical populations with. However, we realize that changing these sentences might have led to confusion, and therefore have undone these changes (p. 5, line 104; p. 8, line 182).

- Page 7, line 143: “expressed as T-scores, with a mean of 50 and a standard deviation of 10 for the US general population [16]”: Please check this reference, as I think it does not refer to scoring interpretation of PROMIS. Should this be reference 17 instead?

The reviewer is correct, this should be reference 17. We have changed this in the manuscript (p. 7, line 140).

- Pages 10 (line 216) and 11 (line 220), reference 58: Given this reference is a personal e-mail sent to the author, I don’t think this can be included among the references of the article.

We have checked the journal requirements and the reviewer is correct. As such, we have deleted this reference from the manuscript (p. 10, lines 204 and 208).

- Page 12, line 248: “The Stocking-Lord constants were Α=0.7604 and Β=-0.1070 for anxiety and Α=0.7786 and Β=0.0339 for depression”: Following my suggestions the information regarding stocking and lord transformation has been substantially reduced, which I think is Good. Now, I don’t think it is relevant to include the SL constants A and B for anxiety and depression. Otherwise, if included, additional indication should be provided to indicate what A and B mean (perhaps keep the formulas).

The reviewer makes a valid remark. We have removed the SL constants from the manuscript (p. 11, lines 236-237).

---

## [Decision Letter · Decision Letter 2]

8 Aug 2022

Towards standardization of measuring anxiety and depression: Differential item functioning for language and Dutch reference values of PROMIS item banks

PONE-D-21-18689R2

Dear Dr. Terwee

We’re pleased to inform you that your manuscript has been judged scientifically suitable for publication and will be formally accepted for publication once it meets all outstanding technical requirements.

Kind regards,

Thiago Machado Ardenghi

Academic Editor

PLOS ONE

Additional Editor Comments (optional):

Reviewers' comments:

Reviewer's Responses to Questions

**Comments to the Author**

1. If the authors have adequately addressed your comments raised in a previous round of review and you feel that this manuscript is now acceptable for publication, you may indicate that here to bypass the “Comments to the Author” section, enter your conflict of interest statement in the “Confidential to Editor” section, and submit your "Accept" recommendation.

Reviewer #1: All comments have been addressed

Reviewer #2: All comments have been addressed

Reviewer #3: (No Response)

2. Is the manuscript technically sound, and do the data support the conclusions?

Reviewer #1: Yes

Reviewer #2: Yes

Reviewer #3: Yes

3. Has the statistical analysis been performed appropriately and rigorously? 

Reviewer #1: Yes

Reviewer #2: Yes

Reviewer #3: Yes

4. Have the authors made all data underlying the findings in their manuscript fully available?

Reviewer #1: Yes

Reviewer #2: Yes

Reviewer #3: Yes

5. Is the manuscript presented in an intelligible fashion and written in standard English?

Reviewer #1: Yes

Reviewer #2: Yes

Reviewer #3: Yes

6. Review Comments to the Author

Reviewer #1: (No Response)

Reviewer #2: The authors have adequately addressed all my comments and I think the article is adequate and ready for publication.

Reviewer #3: The present study assessed DIF for language between the Netherlands and the US for the PROMIS Anxiety and Depression item banks, and presented Dutch reference values for the general population and relevant subpopulations.

I believe the paper satisfies the criteria to be accepted for publication in PLOS one.

The study presents the results of original research. Analyses are performed to a high technical standard and are described in sufficient detail. Conclusions are presented in an appropriate fashion and are supported by the data. Also, the article is presented in an intelligible fashion and is written in standard English.

7. PLOS authors have the option to publish the peer review history of their article (what does this mean?). If published, this will include your full peer review and any attached files.

Reviewer #1: No

Reviewer #2: No

Reviewer #3: **Yes: **Marilia Leão Goettems

---

## [Editor Report · Acceptance letter]

10 Aug 2022

PONE-D-21-18689R2 

Towards standardization of measuring anxiety and depression: Differential item functioning for language and Dutch reference values of PROMIS item banks 

Dear Dr. Terwee:

I'm pleased to inform you that your manuscript has been deemed suitable for publication in PLOS ONE. Congratulations! Your manuscript is now with our production department. 

Kind regards, 

on behalf of

Dr. Thiago Machado Ardenghi 

Academic Editor

PLOS ONE